# Per-Irradiation Monitoring by kV-2D Acquisitions in Stereotactic Treatment of Spinal and Non-Spinal Bony Metastases Using an On-Board Imager of a Linear Accelerator

**DOI:** 10.3390/cancers16244267

**Published:** 2024-12-22

**Authors:** Ahmed Hadj Henni, Geoffrey Martinage, Lucie Lebret, Ilias Arhoun

**Affiliations:** 1Radiation Oncology Department, Centre Frederic Joliot, 76000 Rouen, France; geoffrey.martinage@centrefredericjoliot.fr (G.M.); lucie.lebret@centrefredericjoliot.fr (L.L.); ilias.arhoun@centrefredericjoliot.fr (I.A.); 2Oncology Department, Clinique Saint Hilaire, 76000 Rouen, France

**Keywords:** bone SBRT, image-guided radiotherapy, intra-fraction motion, on-board imager, triggered kV imaging, kV-CBCT

## Abstract

An on-board imager on a linear accelerator allows the acquisition of kV-2D images during irradiation. Overlaying specific structures on these images enables the visual verification of movement at regular frequencies. The aim of this study was to validate the method of the visual tracking of the target volume motion for the stereotactic treatment of bone metastases. To the best of our knowledge, this image-guided radiation therapy (IGRT) method has never been studied in non-spinal bony sites. The results were obtained using measurements from an anthropomorphic phantom and analysis of kV–cone beam computed tomography images from 29 patients treated at our institution. The results validated a visual tracking accuracy of 2.0 mm for spinal sites and 3.0 mm for non-vertebral bone locations. This method, based on an imaging device that is available on current linear accelerators, enables a robust IGRT strategy for performing bone stereotactic treatment at no additional cost to centers.

## 1. Introduction

Stereotactic body radiotherapy (SBRT) is a technique based on delivering high doses of radiation with millimeter precision. However, these treatments are often associated with strong dose gradients and reduced margins to spare the normal tissues. The goal is to achieve a high biologically effective dose (BED) in a limited number of fractions to deliver ablative doses to the target in the treatment of patients with a localized tumor or oligometastasis. Spinal SBRT fits this definition, particularly because of the proximity of nerve structures (the spinal cord, cauda equina, and nerve roots) to the planning target volume (PTV). For this site, several prospective and retrospective studies have shown that a high BED results in local control rates of approximately 80–90% in one year [1,2,3]. The most commonly reported protocols in the literature are as follows: 16–24 Gy in 1 fraction, 24 Gy in 2 fractions, 24–27 Gy in 3 fractions, and 30–35 Gy in 5 fractions [4,5,6]. The complexity of dosimetric planning using intensity modulation to enable a high PTV conformation is correlated with irradiation using adapted immobilization systems and an image-guided radiation therapy (IGRT) strategy. The level of precision required could be of the order of 1 mm/2°, depending on the proximity of the organs at risk [7]. Wang et al. [7] performed a dosimetric study on 20 vertebral metastases and recommended this level of precision to maintain the risk of target volume coverage loss at <5% and the dose increase in organs at risk at <25%.

The appropriate choice of a non-invasive positioning system for the treatment of bone metastases currently guarantees, on average, very low positioning deviations [8]. However, several teams have questioned the relevance of intra-fraction imaging. Indeed, during the stereotactic treatment of bone metastases, the per-beam monitoring of the target movement shows that involuntary patient movements are possible and frequent [9,10].

For example, a study by Hadj Henni et al. [11], using kV-2D images acquired by a system not mounted on a linear accelerator, showed that these offsets could occur at any time during irradiation, with values sometimes exceeding 3 mm. These offsets were corrected at a frequency of ≤1 min and allowed to guarantee a positioning of <1 mm and <2° in all cases.

These results clearly demonstrate that an IGRT approach based solely on three-dimensional cone-beam computed tomography (3D-CBCT) pre- and post-treatment scans is not sufficient to ensure millimeter accuracy in bone SBRT. For the CyberKnife platform (Accuray Inc., Sunnyvale, CA, USA), an imaging system [12] consisting of two X-ray sources and two detectors installed on the floor and ceiling of the treatment room allows for the acquisition of two oblique kV-2D images, and, thus, the real-time monitoring of the target volume position. Linear accelerators, such as the Varian TrueBeam (Varian Medical System, Palo Alto, CA, USA), require the addition of an external kV-2D imaging system to benefit from the same technological capabilities as CyberKnife. The ExacTrac X-Ray 6D orthogonal imaging system fulfills this function perfectly [13], but its cost may not fit the center’s budget. Systems based on tracking the patient’s surface rather than the target volume are more accessible but have not been widely used for the SBRT of bone metastases [14,15]. All other technological options for real-time tumor position tracking require invasive fiducial implantation [16,17].

Varian TrueBeam machines are equipped with an on-board imager (OBI) system that enables the acquisition of 3D CBCT images that can be used to verify patient positioning between each beam of the treatment plan. This intra-fraction IGRT method, often described in the literature [18,19,20,21], increases treatment time, and, therefore, the possibility of patient movement [8]. OBI also offers the possibility of acquiring kV-2D images during irradiation using different triggers (Monitor Unit, degree, or time). In addition, when fiducial markers are implanted, this option (auto-beam hold) allows them to be detected, and the beam to be stopped automatically if they are outside a predefined tolerance zone [22,23]. In the absence of fiducial markers, which is the case in most bone SBRT treatments, only the visual monitoring of these planar images is possible.

Koo et al. [24] were among the first to propose a validated IGRT strategy based on this OBI option for thoracic and lumbar spinal stereotactic treatments. By overlaying the patient’s anatomical structures (vertebral body and spinous process) previously delineated on the planning CT of these kV-2D images, this team was able to detect and correct the intra-fractional motion in 11 of 94 fractions.

The aim of our study was, first, to validate the feasibility of this approach on an anthropomorphic phantom, based on the visual tracking of the target vertebra as well as the inferior and superior vertebrae, using kV-2D imaging during motion irradiation. The choice of a vertebral phantom was motivated by the proximity of the spinal cord to the target volume in this case. Additionally, the effect of operator experience on the detection efficiency of millimeter-scale positional deviations was evaluated.

Clinical feedback from 29 patients validated this approach for guiding irradiation in spinal stereotactic treatment and allowed it to be extended to other extra-vertebral bone sites in a second phase. Structures that facilitate the visual detection of positioning discrepancies in extra-vertebral bone metastases have also been proposed for the clinical use of intra-fraction monitoring with an on-board imager. To the best of our knowledge, this image-guided radiation therapy method has never been studied in non-spinal bony sites.

## 2. Materials and Methods

### 2.1. Phantom Study

Prior to its use in routine clinical practice, a study was performed on a thoracic anthropomorphic phantom developed by RTSafe© (RTsafe P.C., Athens, Greece) (Figure 1) to evaluate the performance of visual tracking of the target volume position. Inspired by Koo et al. [24], offsets of 0.0, 1.0, 1.5, and 2.0 mm were simulated in three translational directions. On CT acquisition (Somatom Definition AS20 RT, Siemens©, Washington, DC, USA) of the phantom with a slice thickness of 1.0 mm, the target, overlying, and underlying vertebrae were delineated to serve as tracking structures of the positional offset in the vertical, longitudinal, and lateral directions. A full-arc treatment plan with X6 FFF (flattening filter-free) beam was simulated using TPS Eclipse (AcurosXB 15.6, 0.1 cm grid, Varian Medical Systems, Palo Alto, CA, USA) to enable experimental manipulation in the treatment room.

In this study, intra-fraction rotational deviations from the reference position were not considered. Numerous studies, such as those by Wang et al. [7] and GukenBerger et al. [25], have defined acceptable rotation thresholds (pitch, yaw, and roll) between 2° and 3.5° without significant dosimetric effects on the spinal cord. For vacuum bag positioning systems, these rotational corrections are well below thresholds [8,11].

For each translational deviation simulated at the accelerator, kV-2D images were acquired every 45° from 0° to 315° with the volumes of interest superimposed (Figure 1). A set of 35 randomly selected images was collected from the set of available images. Without knowing the occurrence and amplitude of the discrepancies, 18 operators (14 radiation therapist, 2 physicians, and 2 physicists) were asked to visually assess these discrepancies and to choose between the answers “Yes, error requiring treatment interruption and kV-CBCT acquisition” and “No, no treatment interruption”.

They were also asked to record their decision-making times. Cases in which the errors were parallel to the acquisition angles of the OBI images could not be detected and were excluded from this test. However, images with an offset of 0.0 were retained. To familiarize users with this blind test, a preliminary self-training step was proposed. This consisted of 12 sample kV-2D images, including information on the acquisition angle, amplitude, and direction of the applied offsets.

The evaluation considered the experience of the operators, who were divided into two groups: referent and non-referent. The referent group comprised professionals directly involved in implementing the IGRT strategy for stereotactic bone treatment at our center. This group included two physicians, one radiation therapist, and one physicist.

### 2.2. Patient Study

The data analyzed in this study were obtained from 29 patients treated at our center under stereotactic conditions for one or more bone metastases (35 tumor sites). The total dose was, in most cases (n = 33), 35 Gy in 5 fractions of 7 Gy (n = 33), delivered on one day out of two. The treatment period ranged from 03/2023 to 03/2024. Table 1 presents the treatment characteristics.

All patients were immobilized in the supine position using a vacuum bag system (Meicen©, Ektelesi Medical, Paris, France) below T3, with the arms raised on an indexed armrest. Above T3, a thermoplastic mask was used with the arms alongside the body. In addition to these devices, the knee and footrest were used. CT images (Somatom Definition AS20 RT, Siemens©) were acquired using a standardized protocol for bone stereotactic treatment, with a slice thickness of 1.0 mm.

For spinal lesions, the gross target volume (GTV) was delineated by experienced physicians after the registration of various imaging modalities (CT, MRI with millimetric slices, and PET-CT when available). The clinical target volume (CTV) was delineated according to the international recommendations of Cox et al. [26]. In the case of epidural lesions, a distance of 3 mm was maintained between the spinal cord and the epidural lesion. For non-vertebral lesions, the CTV corresponded to a geometric extension of 5 mm from the GTV (which could be increased to 10 mm in certain cases) for bone disease and extra-osseous extensions (if extended at this level), respecting the anatomical barriers [27,28]. A 2 mm margin was applied from the CTV to obtain the PTV. This margin can be reduced to 0 mm in cases with proximity to the spinal cord. For non-vertebral bony lesions, the margin was 3 mm (potentially up to 5 mm). A margin of 2 mm was applied around the spinal cord volume and international dose constraints defined for the spinal cord volume were applied to the planning risk volumes of the spinal cord.

As the treatment did not involve the direct participation of the patient, no consent was required. All retrospective analyses were performed using fully anonymized data, in accordance with the ethical standards of our center and the 1964 Declaration of Helsinki.

All treatments were performed using volumetric modulated arc therapy according to our dosimetric protocol based on three coplanar arcs in 6 MV FFF photons at 1400 MU/min with collimator angles of 45°, 315°, and 95°. All treatment plans were calculated using the same planning system (Eclipse Acuros XB 15.6, 0.1 cm grid; Varian Medical Systems).

Irradiation was delivered in a bunker equipped with a TrueBeam 120 MLC (Varian Medical Systems, Palo Alto, CA, USA) and an OBI, which allowed kV-CBCT acquisitions as well as kV-2D images during irradiation. Patients were positioned on a 6D perfect pitch table.

The IGRT procedure involved acquiring a pretreatment kV-CBCT image with the application of the offsets obtained after the patient was placed on the table. The TrueBeam Varian v4.0 Intra-fraction Motion Review (IMR) application was used to acquire kV-2D images at a frequency of 7 s, for an average (standard deviation) of 2646 (740) MU per fraction, which allowed sufficient intra-fraction temporal tracking and decision time. Superimposed structures of the target vertebra and overlying and underlying vertebrae were used as visual guidance volumes for vertebral treatment. In the present study, the method of Koo et al. was extended to non-vertebral bony sites [24]. The initial tracking volumes covered the entire diseased bone and were then progressively adjusted. Borders and edges were then modified according to clinical feedback. Figure 2 summarizes the guidance volumes used at different sites contoured on CT images. Figure 3 shows some examples of kV-2D images acquired during irradiation and analyzed by operators at the treatment station.

The operators were instructed to stop the treatment if they considered correction of the patient’s position necessary. kV-CBCT was then systematically performed to correct the patient’s position, regardless of the actual offsets obtained. Intra-arc and post-treatment kV-CBCT were used to verify the decisions made by radiation therapists.

### 2.3. Analysis Method

Positioning errors were recorded using the Aria Offline Review module (ARIA 15.6; Varian Medical System, Palo Alto, CA, USA) for the lateral, vertical, and longitudinal translations. The kV-CBCT images acquired after the operators stopped the irradiation arc were used to measure real deviations. Without stops during each of the three arcs, the intra-arc and post-treatment kV-CBCT served the same purpose of quantifying the actual offsets.

Five tolerance thresholds (1.0, 1.5, 2.0, 2.5, and 3.0 mm) were selected for data analysis. These thresholds were used to quantify the number of positioning deviations that exceeded the fixed tolerance (true positives (TP)) and were detected by the operators. True negatives (TN) corresponded to cases in which no deviation was visually observed by the radiation therapists and confirmed by subsequent kV-CBCT (intra-arc or post-treatment). False positives (FP) represented stops that were decided by the operator but whose deviations, as determined by kV-CBCT, were below the predefined threshold. Finally, false negatives (FN) corresponded to deviations above the threshold provided by the intra-arc or post-treatment kV-CBCT acquisition, but were not detected by the operators.

We have separated the analysis of vertebral localizations from that of extra-vertebral bone localizations to take account for the different margins used in these two cases.

## 3. Results

### 3.1. Phantom Study

For all 18 participants, the average, minimum, and maximum scores were 81%, 65%, and 94%, respectively, of the 630 responses analyzed, with an average (standard deviation) decision time of 3.0 s (2.3). The referent group achieved 91% (89%, 94%) and 79% (65%, 89%) in the non-referent group. In this test, the detectability of phantom errors was proportional to their amplitudes. For offsets of 0.0, 1.0, 1.5, and 2.0 mm—independent of image incidence (Figure 4)—the results were 77.8%, 92.6%, 90.0%, and 100% for the expert group, and 77.8%, 69.6%, 79.3%, and 88.0% for the less experienced group.

The referent group showed a better ability to detect positioning errors as early as the 1.0 mm threshold, with 100% detectability at 2.0 mm versus 88.0% for the non-referent group. When no offset was applied, both groups did not hesitate to select the answer “Yes, error requiring treatment interruption, and kV-CBCT acquisition” in over 20% of cases. Both groups were aware that omitting positioning errors would be more detrimental to the patient than performing an additional kV-CBCT scan.

The effect of the acquisition incidence was also assessed (Figure 4) for offsets of 1.5 mm and 2.0 mm in the three translation directions. These errors were not detected when the directions were parallel to the OBI angle. In contrast, positioning errors of ≥1.5 mm were 100% detected when the image incidence was perpendicular to the displacement, regardless of the operator’s skill level. When the angle of incidence was intermediate (45°, 135°, or 315°), errors were detected in 94% and 79% of the experts and non-experts, respectively.

### 3.2. Patient Study

In this section, 205 kV-CBCT images were retrospectively reviewed to determine the performance of the proposed IGRT method, which is based on the visual tracking of the target volume position by kV-2D imaging during irradiation with the help of a tumor-specific tracking volume (Figure 2). Of these 205 kV-CBCT, 92 related to vertebral locations and 113 to bony locations outside the vertebrae. Of the 165 available fractions, 135 were used in the present study.

For the vertebral locations and for the three vertical, longitudinal, and lateral directions, the median (maximum) absolute values of the positioning errors were 0.3 (3.6) mm, 0.4 (2.3) mm, and 0.5 (2.1) mm, respectively (Figure 5a). For non-spinal bony metastases and in the same order, the positioning errors were 0.5 (3.3) mm, 0.4 (2.1) mm, and 0.6 (4.8) mm (Figure 5c).

The curves in Figure 5b,d show the proportion (%) of the positioning errors above a given threshold in the three directions. For the vertebral locations, the percentages of errors measured with kV-CBCT images above a particular threshold of 2.0 mm were 3.3%, 2.2%, and 3.3% in the vertical, longitudinal, and lateral directions, respectively. For the bony locations outside the vertebrae, the results for the same threshold and in the same order were 2.7%, 0.9%, and 8.0%. For the latter, taking into account the clinical margins used, the proportion of positioning errors greater than a 3 mm threshold was 0.9%, 0.0%, and 3.5% in the vertical, longitudinal, and lateral directions, respectively.

Table 2 shows the scores obtained by radiation therapists according to different tolerance thresholds using the previously defined IGRT method. The relevance of the decision to stop or not stop the beam during treatment, when the position of the target volume appeared suspicious, was verified using subsequent kV-CBCTs.

FN were the only errors considered harmful to patients and are highlighted in Table 2. FP, which are considered less serious, involve the acquisition of a kV-CBCT of approximately 1.5 min in duration, with an average additional effective dose of 10–20 mSv [29]. For spinal locations and for thresholds of 1.0, 1.5, 2.0, 2.5, and 3.0 mm, the nondetection rates for deviations above these tolerances were 18.5%, 3.3%, 2.2%, 0.0%, and 0.0%, respectively. For non-spinal bony metastases and in the same order, the nondetection rates were 23.0%, 8.0%, 3.5%, 1.8%, and 0.9%.

## 4. Discussion

This study evaluated and validated the IGRT strategy used for the stereotactic treatment of bone metastases, based on the ability to generate periodic kV-2D images during irradiation with superimposed volumes, to facilitate the visual tracking of the target volume position.

The anthropomorphic phantom test confirmed the accuracy of this qualitative target volume tracking method and served as a training tool prior to its implementation in the clinical routine. An analysis of 18 participant responses—which were divided into two groups: referent and non-referent—showed the influence of operator experience on the performance of the proposed IGRT method. Without knowing the occurrence and amplitude of simulated errors of 0.0, 1.0, 1.5, and 2.0 mm in the three translation directions, and without considering the angle of incidence at which the kV-2D images were acquired, the referent group obtained an average correct response of 91% versus 79% for the non-referent group. The larger the offset, the higher the score. For an offset of 2.0 mm, regardless of the angle of image acquisition—except parallel to the direction of the error because it was undetectable—experienced operators achieved 100% success compared with 88% for less experienced operators. These results were significantly different from those reported by Koo et al. [24]. In their study, the detectability rates of the experienced and inexperienced groups were approximately 55% and 28%, respectively, for an offset of 2 mm. This difference can be explained by the experimental conditions and, more specifically, by the type of phantom used. The images of the anthropomorphic Alderson RANDO phantom (Alderson Research Laboratories, Inc., Long Island City, NY, USA) showed artifacts that prevented a good estimation of the simulated shifts. An equally plausible explanation is that the present study included pre-training before phantom testing, which was not the case in the study of Koo et al. [24]. This highlights the importance of training prior to implementing the new IGRT method. Additionally, it highlights the obvious limitations of the results obtained on the phantom compared with the real treatment.

For the experimental protocol used in this study, positioning errors of ≥1.5 mm were detected 100% of the time by all operators when the incidence of the planar image was perpendicular to the displacement. At intermediate angles (45°, 135°, and 315°), the success rate for expert participants was 94%, compared with 74% for the less experienced group. These results indicate that a deviation occurring at an acquisition angle parallel to its direction is likely to be detected at the maximum angle of 45°. With arm rotation speeds in the order of 6°/s, this type of misalignment is detected approximately 7–10 s later, corresponding to one or two new automatically triggered images.

Ong et al. [30] investigated the dosimetric impact of intra-fractional movements on the spinal cord for 6 MV and 6 MV FFF spinal stereotactic treatments. In their study, for 6 MV FFF beams, an offset of 2 mm for 10 s and 30 s induce an increase in Dmax at the spinal cord of 3% and 13%, respectively. The mean (SD) operator-decision time for the present study was 3 (2.3) s. Considering this result and that of Ong et al. [30], the triggering of kV-2D images by the IMR application was set every 7 s during the clinical routine.

During the clinical implementation, we decided to extend the IGRT method beyond the treatment of vertebral bones. The visual guide volume was defined for each site (Figure 2). Clinical cases of patients with orthopedic hardware in the spine are included in this list. Inspired by the study by Cetnar et al. [31], this device was used as a landmark (Figure 2).

The intra-fraction offset values are in the range reported in the literature. Although the majority of offsets were below our clinical threshold of 2 mm and 3 mm depending on the margins applied, our study confirmed the occurrence of sudden movements with amplitudes above these tolerance thresholds.

The rate of deviation of >2 mm and 3 mm, undetected by radiation therapists (FN), was approximately 2% and 1% for spinal locations and for non-spinal bony metastases respectively. These results helped to limit excessive undetected shifts and assumes no distinction between the expertise of the people and the incidence of the planar image in relation to the direction of movement. This result must also be weighed against the fact that movement may occur between the beam stop and kV-CBCT acquisition.

The results also showed that, for a tolerance threshold of 1.0 mm, the detectability (FN) decreased to 81.5% for spinal sites. This performance has not been validated in the clinical routine at our institution and does not allow for the margin reduction of the positioning uncertainty in this threshold. This finding and the limitations of this strategy should be compared with the performance of non-embedded accelerator systems with automatic registration during irradiation, which can guarantee an accuracy of the order of 1.0 mm [32] for the spinal treatment.

We have extended the methodology to non-vertebral bone locations, initially proposing guide volumes that cover the entire diseased bone and then progressively adjusting the boundaries and edges according to clinical feedback. Each team implementing our methodology must use its own clinical feedback to find the most relevant guide volumes. Several image parameters, such as kV, mAs, or filters selected by the operators at the treatment console, can impact the performance of this IGRT strategy. This impact has not been studied in this manuscript. Nevertheless, the results in Table 2 showed that, for spinal sites, eight offsets greater than 2 mm were corrected (true positives) by kV-CBCT acquired immediately after stopping treatment. No CBCT at the end of the arc (false negatives) was greater than 2.5 mm. For non-vertebral bone locations, four offsets greater than 3.0 mm were corrected and only one kV-CBCT at the end of the arc was greater than 3.0 mm.

The tracking volumes shown in Figure 2 were manually delineated, adding a time-consuming step to the treatment plan. This was compounded by the fact that the present study was performed using a CT scanner with a slice thickness of 1.0 mm. The effect of CT slice thickness on the performance of this IGRT method has been investigated by several authors [33]. For example, Koo et al. [24] reported a decrease in detectability of >10% for a 2 mm threshold when comparing the 1.0 mm and 3.0 mm CT scans. Therefore, to maintain the accuracy of the IGRT method, the thickness of the dosimetric scans was not increased. Automatic segmentation strategies reduce the time required for this task while harmonizing the delineation of the positioning structures.

Notably, this method is expertise-dependent and requires maximum operator concentration in the workplace. Therefore, training is a key factor in the effectiveness of monitoring qualitative target volume positions. At least two radiation therapists supervised the irradiation fraction and 2D kV images. To limit interruptions, we established a procedure to clearly signal the progress of stereotactic bone treatment. During these treatments, only those directly involved were present, a “stereotactic treatment in progress” flag was installed, and telephone calls were turned off.

Automated beam stopping is possible with this type of machine [23], but it requires the implantation of fiducials, as in the case of prostate cancer treatment. A major step forward in the management of motion in bone SBRT is the development of an automated beam stop by the manufacturer, which should be available with the IMR application and based on bone registration. Currently, the only alternative is external imaging devices based on bone registration [13].

To date, many centers use kV-CBCT imaging only before the start of each arc, and, at best, between each arc. However, this adds time to the treatment fraction and delays the detection of patient motion. TPS Eclipse can provide the exact timing of the stops that occur during the irradiation arcs. This information will allow the investigation to continue by estimating the dosimetric impact on the target volume and OAR according to the IGRT strategy used, with or without kV-2D imaging during irradiation.

## 5. Conclusions

The use of OBI kV-2D imaging for the visual guidance of spinal and non-spinal bony stereotactic radiation was validated using anthropomorphic phantom and clinical data. The results confirmed a visual accuracy of 2 mm for spinal stereotactic treatment and 3 mm for non-spinal stereotactic treatment. These results were consistent with the positioning uncertainty margins used. The study also proposed structures to guide the visual monitoring of the target volume position for different treated sites. The proposed method, based on an imaging device that is always available on current linear accelerators, enables a robust IGRT strategy for performing bone stereotactic treatment at no additional cost to centers.

## Figures and Tables

**Figure 1 cancers-16-04267-f001:**
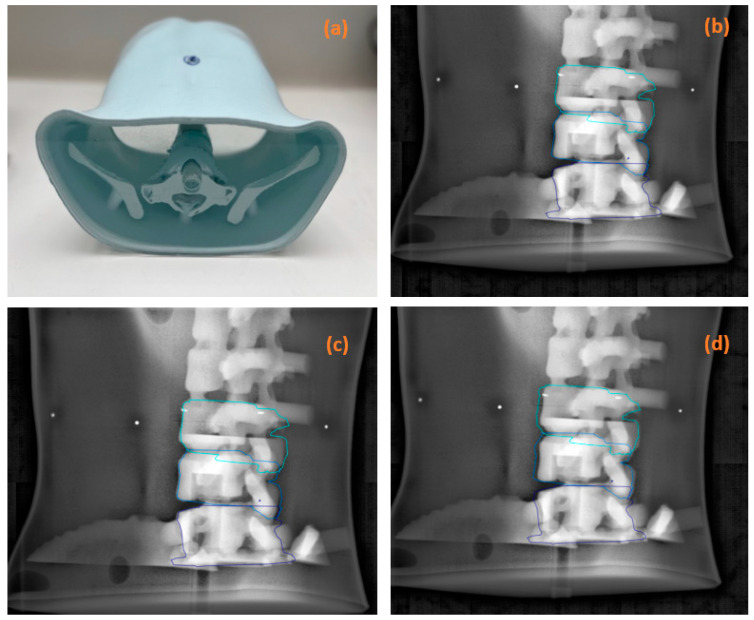
(**a**) RTSafe Spine© phantom. (**b**–**d**) Examples of kV-2D images analyzed by 18 participants during the anthropomorphic phantom test. The angle of incidence of the images was 135°, with deviations of (**b**) 0.0 mm, (**c**) 2.0 mm laterally, and (**d**) 2.0 mm vertically. Visual guidance structures were superimposed onto the images.

**Figure 2 cancers-16-04267-f002:**
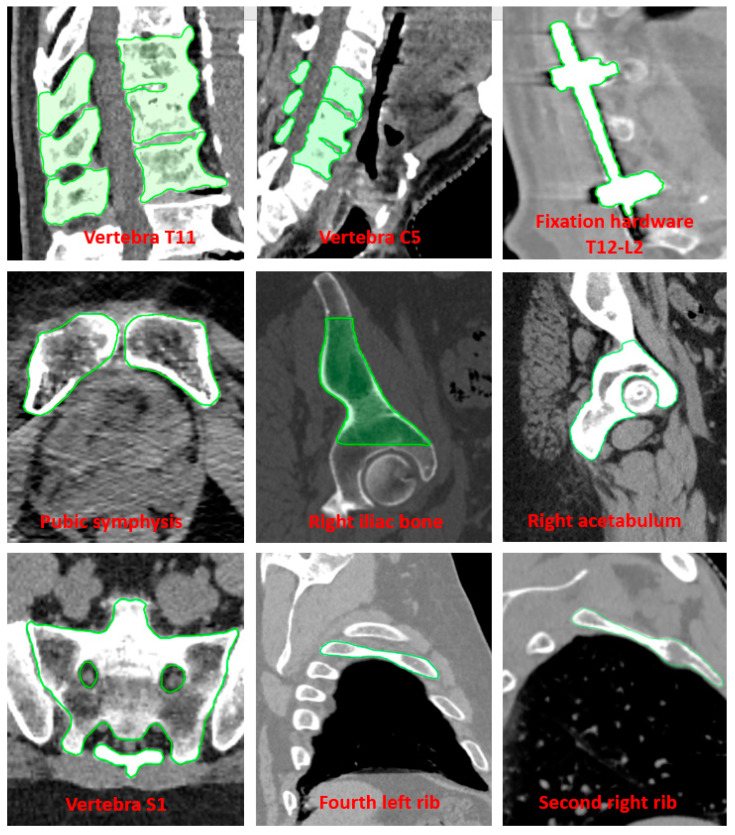
Examples of delineated structure types on millimeter-slice-thickness CT used for bone stereotactic IGRT based on kV acquisition during irradiation.

**Figure 3 cancers-16-04267-f003:**
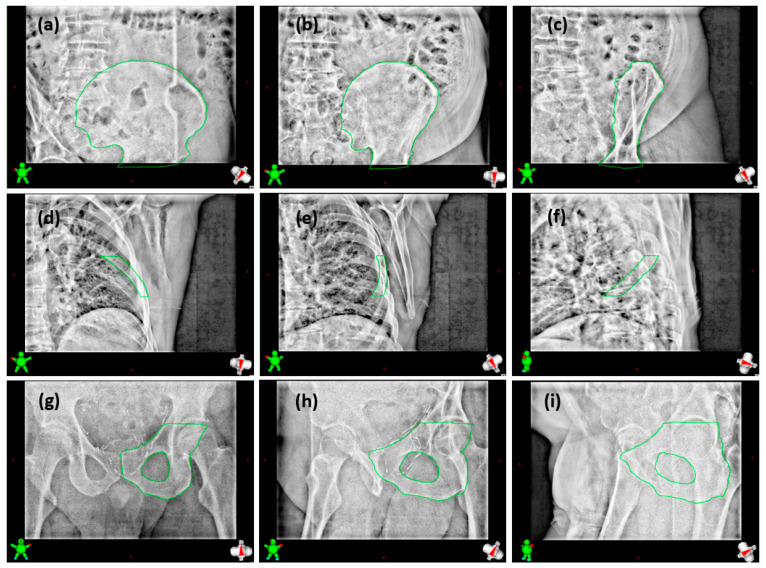
Examples of kV-2D images acquired during treatment fractions: (**a**–**c**) images of a right iliac wing acquired at 215.7°, 183.8°, and 147.9°; (**d**–**f**) images of a right fifth rib acquired at 188.3°, 147.7°, and 105.9°; and (**g**–**i**) images of a left ischium acquired at 359.3°, 29.4°, and 59.4°.

**Figure 4 cancers-16-04267-f004:**
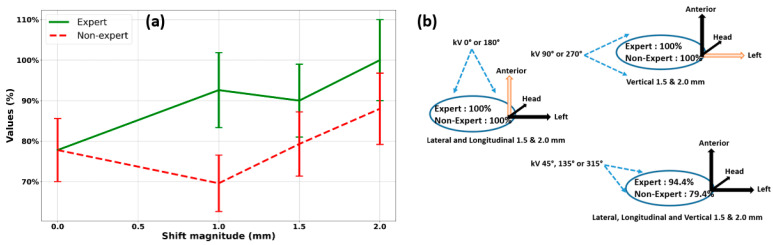
(**a**) Proportion (%) of correct responses to the anthropomorphic phantom test according to the amplitude of the shifts applied for the referent group (green solid line) and the less experienced group (red dashed line). (**b**) Evaluation of the acquisition incidence effect for offsets of 1.5 mm and 2.0 mm in the three translation directions.

**Figure 5 cancers-16-04267-f005:**
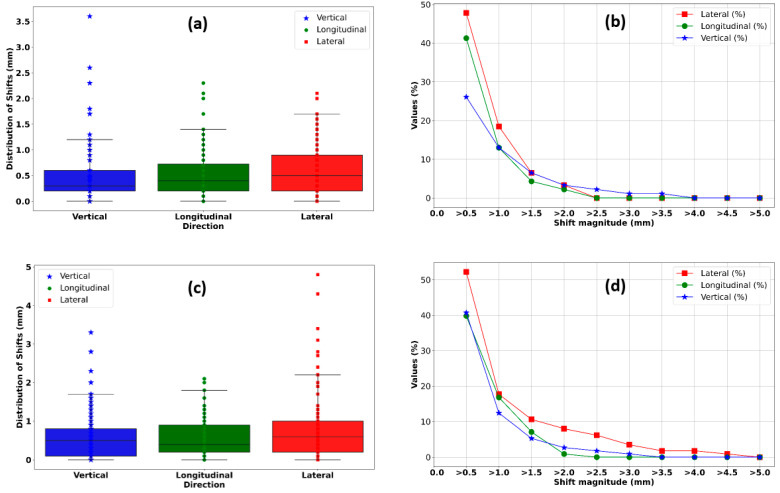
(**a**,**c**) Distribution of observed deviations for the three directions: vertical, longitudinal, and lateral for spinal locations and for non-spinal bony metastases, respectively. (**b**,**d**) Proportion (%) of observed deviations by direction and above a given threshold in millimeters for spinal locations and for non-spinal bony metastases, respectively.

**Table 1 cancers-16-04267-t001:** Patient and treatment characteristics. Abbreviations: ADK = adenocarcinoma; CCI = carcinoma; NET = neuroendocrine tumor; CCRCC = clear cell renal cell carcinoma; NA = not applicable; WHO = World Health Organization.

Patients	Site Treated	WHO Performance Status	Primary Cancer	Histological Types	Sex	21 Men
						8 Women
					Median Age (Years) (Range)	71 (47–85)
					Dose/Fractions	
1	Sacrum	0	Prostate	ADK	35 Gy/5 fr	
2	Right iliac wing	0	Prostate	ADK	35 Gy/5 fr	
3	Acromion	0	Breast	CCI	35 Gy/5 fr	
4	Right 10th rib	0	Prostate	ADK	35 Gy/5 fr	
5	Pubic symphysis	1	Breast	CCI	35 Gy/5 fr	
6	Right ischium	NA	Prostate	ADK	30 Gy/6 fr	
7	L5-S1	0	Prostate	ADK	36 Gy/6 fr	
8	T5	0	Prostate	ADK	35 Gy/5 fr	
9	Right iliac wing	0	Prostate	ADK	35 Gy/5 fr	
10	L5	NA	Breast	ADK	35 Gy/5 fr	
11	First right rib	0	Prostate	ADK	35 Gy/5 fr	
	Right 7th rib	0	Prostate	ADK	35 Gy/5 fr	
12	Left 10th rib	0	Kydney	CCRCC	35 Gy/5 fr	
13	Right ischium	0	Prostate	ADK	35 Gy/5 fr	
14	L4	0	Prostate	ADK	35 Gy/5 fr	
15	T12	1	Prostate	ADK	35 Gy/5 fr	
	Left 5th rib	1	Prostate	ADK	35 Gy/5 fr	
16	T12-L2 post-operative	NA	Breast	CCI	30 Gy/10 fr	
17	Right iliac wing	0	Prostate	ADK	35 Gy/5 fr	
	Right 6th rib	0	Prostate	ADK	35 Gy/5 fr	
	Left 6th rib	0	Prostate	ADK	35 Gy/5 fr	
18	T12	0	Breast	ADK	35 Gy/5 fr	
19	Right ischium	0	Prostate	ADK	35 Gy/5 fr	
20	T12	0	Prostate	ADK	35 Gy/5 fr	
21	Right posterior iliac wing	0	Prostate	ADK	35 Gy/5 fr	
	Right anterior iliac wing	0	Prostate	ADK	35 Gy/5 fr	
	T3	0	Prostate	ADK	35 Gy/5 fr	
22	L1	0	Breast	ADK	35 Gy/5 fr	
23	T12	0	Kidney	NET	35 Gy/5 fr	
24	C5	0	Prostate	ADK	35 Gy/5 fr	
25	Left ischium	0	Breast	ADK	35 Gy/5 fr	
26	T12	0	Breast	ADK	35 Gy/5 fr	
27	T11	1	Prostate	ADK	35 Gy/5 fr	
28	T2	0	Prostate	ADK	35 Gy/5 fr	
29	T1	0	Prostate	ADK	35 Gy/5 fr	

**Table 2 cancers-16-04267-t002:** Proportions of true positives (TP), true negatives (TN), false positives (FP), and false negatives (FN) according to the tolerance thresholds applied to the analysis of the operator’s decisions in the treatment room. FN are shown in bold. Abbreviations: VB = vertebrae; NVB = non-vertebral bone.

92 and 113 kV-CBCT Scenarios for VB and NVB Sites, Respectively	Designation		Threshold				
			1.0 mm	1.5 mm	2.0 mm	2.5 mm	3.0 mm
Beam stop and subsequent kV-CBCT acquisition ≥ threshold	True Positives (TP)	VB	21 (22.8%)	14 (15.2%)	8 (8.7%) 11 (9.7%)	2 (2.2%) 11 (9.7%)	1 (1.1%) 4 (3.5%)
		NVB	31 (27.4%)	14 (12.4%)			
No beam stop and kV-CBCT acquisition at end of arc < threshold	True Negatives (TN)	VB	42 (45.6%)	56 (60.9%)	57 (62.0%)	59 (64.1%)	59 (64.1%)
		NVB	45 (39.8%)	62 (54.9%)	67 (59.3%)	69 (61.1%)	70 (61.9%)
Beam stop and subsequent kV-CBCT acquisition < threshold	False Positives (FP)	VB	12 (13.0%)	19 (20.6%) 28 (24.8%)	25 (27.2%) 31 (27.4%)	31 (33.7%) 35 (31.0%)	32 (34.8%) 38 (33.6%)
		NVB	11 (9.7%)				
**No beam stop and kV-CBCT acquisition at end of arc ≥ threshold**	**False Negatives (FN)**	**VB**	**17 (18.5%)**	**3 (3.3%)** **9 (8.0%)**	**2 (2.2%)** **4 (3.5%)**	**0 (0.0%)** **2 (1.8%)**	**0 (0.0%)** **1 (0.9%)**
		**NVB**	**26 (23.0%)**				

## Data Availability

The original contributions of this study are included in the article. Further inquiries can be directed to the corresponding authors.

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
