# Peer review of "Per-Irradiation Monitoring by kV-2D Acquisitions in Stereotactic Treatment of Spinal and Non-Spinal Bony Metastases Using an On-Board Imager of a Linear Accelerator"

_cancers, 2024, doi:10.3390/cancers16244267_

Round 1
Reviewer 1 Report
Comments and Suggestions for Authors
The authors present an interesting study on a topic of relevance to many treatment centers. The study was similar to previous studies, particularly Koo et al, with the main difference being the inclusion of non-spinal bone targets. The authors should place more emphasis on areas where their study builds on the previous study including discussion of the importance of these new contributions.
I don’t think it makes sense to remove the shifts that are parallel to the kV direction from your analysis. These are cases where your method failed to detect movement, so should be considered as a “false negative”. Looking at Koo et al, they do not appear to remove these cases from their analysis, which is likely part of the reason why your results seem much better than theirs.
The main difference between this study and Koo is that non-VB targets were used in this study. However, the data is presented with both VB and non-VB bony targets lumped together. It would be helpful to see these separated so that conclusions can be made specifically about non-VB targets.
Title and throughout: “non-spinal metastases” implies the inclusion of soft tissue targets, whereas only bone targets are included in this study. The fact that these are bone targets should be clear (e.g. “non-spinal bony metastases” or “non-spinal metastases to the bone”)
It would be helpful to see some sample kV images with overlaid contours, including some where shifts were detected and others where they were not.
If possible, include data on how often a shift was caught on a subsequent kV image when it was missed on a previous kV (or perhaps how long until an error is caught after a shift was missed). Part of the benefit of the way this technique is implemented is that images are acquired often, minimizing the time that a shift goes undetected even when it is undetectable at some angles. Having a better idea of how long it takes to catch a shift would help highlight this advantage.
Figure 3b: Why is the “head” direction for 90 and 270 kV not considered?
Line 46: “treatment of patients with oligometastasis” – This seems to imply that SBRT is not ever used for primary disease.
98-103: This paragraph was somewhat confusing. They begin by saying that their area of novelty is the addition of non-spinal sites. Then say that “therefore” they tested their method in a VB phantom. It leaves the reader wondering why they didn’t use a non-spinal phantom if that is where the main contribution lies.
Conclusion (lines 373-374): “The study also proposed structures to guide the visual monitoring of the target volume position for different treated sites.” More discussion on how to choose a non-spinal overlay target is needed to include this statement in the conclusion.
Author Response
Author response: Dear reviewer, thank you for your answers and the associated comments to improve our manuscript. We have responded to each of these points and incorporated them in our manuscript.
The authors present an interesting study on a topic of relevance to many treatment centers. The study was similar to previous studies, particularly Koo et al, with the main difference being the inclusion of non-spinal bone targets. The authors should place more emphasis on areas where their study builds on the previous study including discussion of the importance of these new contributions.
Author response: Thank you for your comments. This new version focuses more on the contribution of our work. As suggested in the comments, we have separated the analysis of vertebral localisations from that of extra-vertebral localisations (Table 2 and Figure 5). This has allowed us to emphasise the latter in different parts of the manuscript.
Given the differences observed in relation to the study by Koo et al, we felt it was important to highlight the significant differences between the two studies. For this reason, we also carried out a detailed analysis of the spinal locations in order to provide a more comprehensive comparison and to highlight the specific features of our approach.
I don’t think it makes sense to remove the shifts that are parallel to the kV direction from your analysis. These are cases where your method failed to detect movement, so should be considered as a “false negative”. Looking at Koo et al, they do not appear to remove these cases from their analysis, which is likely part of the reason why your results seem much better than theirs.
Author response: We have not excluded any cases from our analysis. In the phantom study, it is impossible to distinguish between motion parallel to kV acquisition and no motion. We have included images with no offset in our evaluation. The correct answer in this case is “No, no treatment interruption”. For patient analysis, systematic CBCT between each arc, after an operator stop and at the end of the session enabled us to distinguish VP, FP, VN and FN. This methodology, with the limitations discussed in more detail in the discussion section, allowed us to measure the effectiveness of our IGRT strategy.
The main difference between this study and Koo is that non-VB targets were used in this study. However, the data is presented with both VB and non-VB bony targets lumped together. It would be helpful to see these separated so that conclusions can be made specifically about non-VB targets.
Author response: Yes, we've taken your comment fully into account by modifying Table 2, Figure 5 and the text in different part of the document.
Title and throughout: “non-spinal metastases” implies the inclusion of soft tissue targets, whereas only bone targets are included in this study. The fact that these are bone targets should be clear (e.g. “non-spinal bony metastases” or “non-spinal metastases to the bone”)
Author response: Yes, you're right, we've modified this expression wherever necessary in the manuscript.
It would be helpful to see some sample kV images with overlaid contours, including some where shifts were detected and others where they were not.
Author response: We've added a figure (figure 3) to the document showing examples of kV images acquired during irradiation in stereotactic extra-vertebral bone treatment fractions. The images are taken from screenshots at the treatment console. Offline review on the Aria record and verify only records kV images without superimposed contours. The cases presented did not require any stops. Our manipulators considered that the bone limits did not extend beyond the visual tracking guide contours. It is then up to each team to assess the accuracy of this qualitative follow-up, using the methodology proposed here. In our case, it allowed us to correct offsets > 2mm (true positives) 8 times for vertebral locations and 11 times for non-vertebral bone locations. We have highlighted the positive contribution of this technique in the discussion section.
If possible, include data on how often a shift was caught on a subsequent kV image when it was missed on a previous kV (or perhaps how long until an error is caught after a shift was missed). Part of the benefit of the way this technique is implemented is that images are acquired often, minimizing the time that a shift goes undetected even when it is undetectable at some angles. Having a better idea of how long it takes to catch a shift would help highlight this advantage.
Author response: We have presented the results (now Figure 4) of testing the different operators according to the angle of acquisition of the kV-2D. In the discussion section (lines 324-337), we addressed this point by linking it to the study by Ong et al [30] on the dosimetric impact of sudden movements according to their duration. It is difficult to know the exact moment when a movement occurs, especially when it occurs in parallel with the kV-2D acquisition. However, as you point out, the acquisition frequency minimises the time during which this error is undetectable. A deviation of 2 mm for 10 s in 6 MV FFF increases the Dmax at the spinal cord by 3%. In response to your comment, we have clarified this limitation of our study.
Figure 3b: Why is the “head” direction for 90 and 270 kV not considered?
Author response: Thank you for your comment. It was indeed an error on our part, which has been corrected.
Line 46: “treatment of patients with oligometastasis” – This seems to imply that SBRT is not ever used for primary disease.
Author response: Yes, we've changed that sentence too.
98-103: This paragraph was somewhat confusing. They begin by saying that their area of novelty is the addition of non-spinal sites. Then say that “therefore” they tested their method in a VB phantom. It leaves the reader wondering why they didn’t use a non-spinal phantom if that is where the main contribution lies.
Author response: We chose to use a spinal phantom because of its proximity to the spinal cord, which is a critical structure and a major challenge in radiotherapy. Although our main contribution lies in the addition of non-spinal sites, the spinal phantom was initially used to validate the methodology in a particularly restrictive context. This approach allowed us to ensure that our method could be adapted to the most complex cases before extending it to non-spinal sites. For these sites, we defined guide volumes covering the entire affected bone, and then progressively adjusted the boundaries and edges according to clinical feedback. Finally, kV-CBCT images were taken at each key stage, in parallel with kV-2D images taken during irradiation, allowing precise verification of positioning. We have modified this paragraph to explain our approach more clearly.
Conclusion (lines 373-374): “The study also proposed structures to guide the visual monitoring of the target volume position for different treated sites.” More discussion on how to choose a non-spinal overlay target is needed to include this statement in the conclusion.
Author response: Yes, we have added a paragraph in the materials and methods section to explain how to delineate our guide volumes.

Reviewer 2 Report
Comments and Suggestions for Authors
The current study investigates the clinical use of planar OBI and superimposed structures for visual image guidance in bone stereotactic treatment. Validation was performed using both anthropomorphic phantoms and patient treatments, which provides a foundation for the study's findings. However, several key aspects require clarification and enhancement to strengthen the manuscript.
First, the study would benefit from a clearer explanation of the statistical analysis used to assess the significance of interobserver differences, particularly in Figure 3. The absence of this detail leaves the interpretation of the results incomplete. Additionally, there is concern about the subjectivity of the studies. It would be helpful to address these concerns by providing more objective measures or justifying the methodological choices.
The discussion section should be expanded to include the limitations and strengths of the study, which are currently underrepresented. Furthermore, the diagnosis of the current cohort of patients should be explicitly stated to provide context for the clinical relevance of the findings. It would also be valuable to explore factors potentially associated with worse simulations, such as patient age, KPS, or other relevant clinical variables. These considerations could offer deeper insights into the study's outcomes.
The current study investigates the clinical use of planar on-board imaging and superimposed structures for visual image guidance in bone stereotactic treatment. Validation was performed using both anthropomorphic phantoms and patient treatments, which provides a foundation for the study's findings. However, several key aspects require clarification and enhancement to strengthen the manuscript.
First, the study would benefit from a clearer explanation of the statistical analysis used to assess the significance of interobserver differences, particularly in Figure 3. The absence of this detail leaves the interpretation of the results incomplete. Additionally, there is concern about the subjectivity of the studies. It would be helpful to address these concerns by providing more objective measures or justifying the methodological choices.
The discussion section should be expanded to include the limitations and strengths of the study, which are currently underrepresented. Furthermore, the diagnosis of the current cohort of patients should be explicitly stated to provide context for the clinical relevance of the findings. It would also be valuable to explore factors potentially associated with worse simulations, such as patient age, Karnofsky Performance Status (KPS), or other relevant clinical variables. These considerations could offer deeper insights into the study's outcomes.
Finally, certain methodological details require clarification, including the inclusion and exclusion criteria used for patient selection. Minor corrections include the redundant spelling out of "IGRT" in line 98. Addressing these comments will enhance the clarity, depth, and impact of the manuscript.
Author Response
The current study investigates the clinical use of planar OBI and superimposed structures for visual image guidance in bone stereotactic treatment. Validation was performed using both anthropomorphic phantoms and patient treatments, which provides a foundation for the study's findings. However, several key aspects require clarification and enhancement to strengthen the manuscript.
Author response: Dear reviewer, thank you for your answers and the associated comments to improve our manuscript. We have responded to each of these points and incorporated them in our manuscript.
First, the study would benefit from a clearer explanation of the statistical analysis used to assess the significance of interobserver differences, particularly in Figure 3. The absence of this detail leaves the interpretation of the results incomplete. Additionally, there is concern about the subjectivity of the studies. It would be helpful to address these concerns by providing more objective measures or justifying the methodological choices.
Author response: The first part of our phantom-based work consisted of reproducing the study by Koo et al [24] to assess the feasibility of this imaging strategy in the case of spinal stereotactic treatment. We preferred to use a vertebral phantom because of its proximity to the spinal cord which is a critical structure and a major challenge in radiotherapy. We included more participants, including radiation therapists (14), physicists (2), and physicians (2).
- Regarding the analysis of differences between observers (Figure 4, formerly Figure 3), we did not perform any statistical tests. However, in order to better interpret the results, we divided the operators into two different groups:
- The expert panel consists of four participants with significant clinical experience and technical investment.
- The non-expert group, consisting of 14 participants, had a more limited involvement in the implementation of the technique.
This differentiation has highlighted potential differences linked to operators' experience and technical expertise.
- We recognize the potential limitations associated with the subjectivity of results in this type of study. To minimize this effect, we included a diverse group of participants and standardized the methodology used by all operators. In addition, clinical feedback was used to refine the limits of the guide volumes. We have modified our manuscript to further justify these methodological decisions.
- Finally, during our discussion, we emphasized the importance of kV-CBCTs taken at each key moment of irradiation. These acquisitions made it possible to validate in real time the significant offsets detected by the KV-2D, thus limiting the risk of error.
To clarify our methodology, we have differentiated the results obtained for vertebral and non-vertebral bone locations (Figure 5 and Table 2). We have also taken your comments into account by modifying the text in different sections of the manuscript.
The discussion section should be expanded to include the limitations and strengths of the study, which are currently underrepresented. Furthermore, the diagnosis of the current cohort of patients should be explicitly stated to provide context for the clinical relevance of the findings. It would also be valuable to explore factors potentially associated with worse simulations, such as patient age, KPS, or other relevant clinical variables. These considerations could offer deeper insights into the study's outcomes.
Author response: Yes, we changed the discussion section to put more emphasis on the non-vertebral part, which is one of the strengths of our study. We also clarified and added the limitations of our work. The aim of the study is not to determine the potential medical factors of intra-fraction movements, but to detect them when they occur. However, to address your comment, we have provided more details about the type of sites studied in Table 1.
Finally, certain methodological details require clarification, including the inclusion and exclusion criteria used for patient selection. Minor corrections include the redundant spelling out of "IGRT" in line 98. Addressing these comments will enhance the clarity, depth, and impact of the manuscript.
Author response: We performed a retrospective analysis of data from the first 29 patients treated with bone stereotactic treatment at our center without specific inclusion or exclusion criteria. We have modified text line 98

Round 2
Reviewer 2 Report
Comments and Suggestions for Authors
I am satisfied with the point-to-point responses. no more comments.